# ZEST: ZEROSHOT SPARSE FINE-TUNING

## ABSTRACT

Recent studies have pointed out that fine-tuning a subset of layers from the model can match or even outperform the performance of full fine-tuning, known as surgical fine-tuning (Lee et al., 2022). This method effectively helps reduce the risks of overfitting and accelerates the fine-tuning process. However, swiftly and accurately identifying the "right" layers is not straightforward. Existing approaches naively train each layer until convergence and find the best candidates, which is not scalable, especially given the rapid growth in model sizes. In this paper, we propose **ZEST**: **Ze**roshot **S**parse fine-**T**uning. We first study and compare the zero-shot metrics acquired from a single forward and backward pass. We observe that the metrics are inconsistent for different model and dataset combinations, thus we train a universal ZEST predictor to generalize this method. We use the zero-shot ZEST predictor to rank layers by the estimated importance and fine-tune only the important parameters. By doing so, we can decrease the number of trainable parameters by up to 99%, being on par or outperforming full fine-tuning in terms of model performance. We thoroughly evaluate the effectiveness of ZEST on various tasks and modalities. We train a universal predictor for ResNet50, MobilenetV2, and EfficientNet on 8 different datasets. We also scale this method up to BERT and LLAMA. Our results demonstrate that fine-tuning just five layers can closely match or even outperform the performance achieved through full fine-tuning on LLaMA-7B. Specifically, fine-tuning only the **5** fully connected layers on LLaMA chosen by ZEST can result in improvements of up to 5% over full fine-tuning

## 1 INTRODUCTION

Deep learning has revolutionized the field of artificial intelligence, achieving remarkable breakthroughs in various domains. These models are often pre-trained on large annotated data and then fine-tuned on a relatively small dataset. This approach allows the models to better adapt to real-world applications while retaining the knowledge gained during pre-training. Collecting and fine-tuning small labeled datasets can improve downstream performance in a cost-effective manner while substantially outperforming domain generalization and unsupervised adaptation methods (Rosenfeld et al., 2022; Kirichenko et al., 2023). This paradigm has been widely adopted in various tasks (Ren et al., 2015; Kirillov et al., 2023).

Recently, researchers have shown that fine-tuning a small contiguous subset of the entire model can outperform fine-tuning the entire model (Lee et al., 2022). By fine-tuning only a few layers, not only can the number of parameters to be fine-tuned be reduced, but also the overall throughput of the model's forward process can be increased. However, identifying the "important" parameters that are worth fine-tuning is not a trivial task. Existing studies either rely on efficiency priors, such as the bias terms (Houlsby et al., 2019; Cai et al., 2020), or aggressively freeze the front layers. Surgical fine-tuning (Lee et al., 2022) and sparse update (Lin et al., 2022) empirically analyzing the importance of each layer. However, these approaches either suffer from performance degradation or require expensive computational resources. For example, Lin et al. (2022) iterate all layers $l_i \in \{l_1 \dots l_n\}$ in the neural network, freeze all other layer $(\{l_1 \dots l_n\} \backslash \{l_i, \text{classification}\})$, and then train the selected layers. Though the validation accuracy can reveal the layers' contribution, the process requires to **repeat** $n$ **times**: once for each layer and each dataset. Therefore, the method is only feasible for small models on small datasets.

To tackle the challenges, we propose a method called ZEST: Zero-shot Sparse fine-Tuning. With ZEST, we initially perform a contribution analysis of each layer and gather zero-shot metrics during

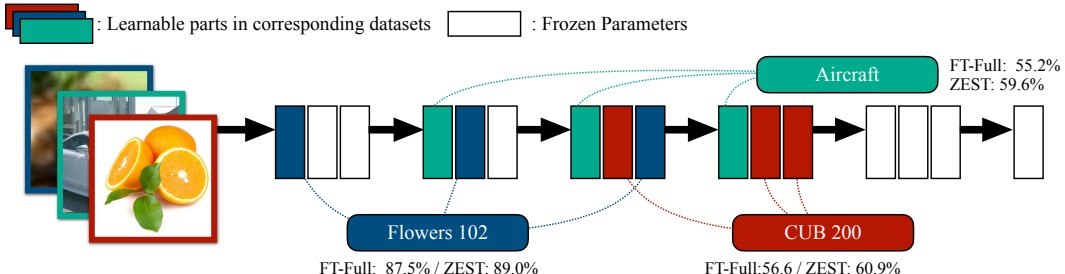

**Figure 1.** The overview of sparse fine-tuning. In transfer learning, it is sometimes more effective to only fine-tune a few blocks of parameters and freeze the remaining parameters, rather than performing full fine-tuning.

training. By analyzing these contribution scores and zero-shot metrics, we discover that while the zero-shot metrics offer some indication of the effectiveness of each layer, they are not applicable across different models and datasets. Consequently, we utilize these zero-shot scores to train the ZEST predictor that can rank the layers on previously unseen datasets and models. We thoroughly evaluate ZEST's effectiveness on both vision and language tasks and show that ZEST can efficiently and accurately find important layers for a new incoming dataset. Our results demonstrate that sifting layers by ZEST can not only improve the final performance but also speed up fine-tuning by 2.0x and reduce memory footprint 2.7x.

In this paper, our contributions are as follows:

- We propose ZEST, a method to automatically find "important parameters" for fine-tuning new datasets. Notably, our ZEST only requires a single round of forward and backward to estimate which is $1000\times$ more efficient than previous methods.

- With ZEST selected layers, fine-tuning only a small percentage of the full parameters (0.3% for LLMs, and $5\%$ to $10\%$ of parameters for CNNs) the accuracy can closely match, or even outperform the full fine-tuning baselines while accelerating the fine-tuning process by back-propagating less layers (1.9x speedup and 2.0 memory saving for MobilenetV2, 2.3x speedup and 2.7x memory saving on LLaMA).

- ZEST does not modify the model architecture nor does it increase the input such as prompt length, thus is orthogonal to existing methods such as LoRA and prefix-tuning. They can be further combined together to further boost performance or further reduction in storage and memory.

We show that ZEST is effective on various models and tasks, and has a large practical utility in fine-tuning models in memory- and compute-constrained environments.

## 2 RELATED WORK

### 2.1 TRANSFER LEARNING

Transfer learning harnesses pre-trained features to enhance a model's performance on tasks and domains that are related but distinct (Oquab et al., 2014; Yosinski et al., 2014; Sharif Razavian et al., 2014). This strategy proves particularly valuable when obtaining labeled data for the target task is constrained or costly. It has gained widespread adoption in various domains, including computer vision tasks such as classification, detection, and segmentation, as well as language-related tasks like translation, summarization, and question answering. Numerous transfer learning methods introduce techniques to regulate the fine-tuning process, with the aim of preserving acquired knowledge and enhancing fine-tuning performance. These methods include works by (Zhang et al., 2020; Xuhong et al., 2018; Lee et al., 2019a; Jiang et al., 2020; Li et al., 2020; Aghajanyan et al., 2020; Gouk et al., 2021; Shen et al., 2021; Karani et al., 2021). In particular, the concept of Module Criticality involves an examination of the loss surface for each layer (Zhang et al., 2019; Chatterji et al., 2019; Neyshabur et al., 2020). Additionally, Surgical Fine-tuning, as proposed by (Lee et al., 2022), recommends strategies such as freezing later layers or employing surgical fine-tuning on earlier layers to potentially achieve superior results in specific scenarios. Drawing inspiration from these insights, our focus lies

in the efficient identification of the optimal subset of layers for fine-tuning, thereby enhancing the fine-tuning process.

## 2.2 PARAMETER FREEZING AND EFFICIENT FINE-TUNING

Parameter freezing is a widely adopted technique in the fine-tuning process, employed in various contexts such as domain adaptation (Sener et al., 2016), early stopping (Mahsereci et al., 2017), generative models (Mo et al., 2020), and gradient-based meta-learning (Zintgraf et al., 2019; Raghu et al., 2019; Triantafillou et al., 2021). Many studies have demonstrated that freezing specific parameters within a pre-trained model can effectively mitigate overfitting during the fine-tuning process (Kirkpatrick et al., 2017; Lee et al., 2019b; Guo et al., 2019; Ramasesh et al., 2020; Liu et al., 2021; Royer & Lampert, 2020; Eastwood et al., 2021; Evci et al., 2022; Eastwood et al., 2022; Cohen et al., 2022; Touvron et al., 2022).

In the age of large foundational models, a series of parameter-efficient fine-tuning (PEFT) techniques have emerged in both vision (Cai et al., 2020; Lin et al., 2020) and language domains (Zaken et al., 2021; Li & Liang, 2021; Hu et al., 2021; Houlsby et al., 2019; Liu et al., 2022). These approaches often advocate for fine-tuning only a subset of the model's parameters, introducing new layers instead of updating existing weights, or optimizing input word embeddings. These modifications aim to preserve the majority of pre-trained parameters, leaving them frozen and unchanged during the fine-tuning process. These practices can be seen as specific instances of sparse fine-tuning, and our experiments underscore the significance of fine-tuning these pivotal parameters.

## 2.3 ZERO-SHOT METRICS AND ESTIMATIONS

Zero-shot metrics refer to the data extracted from a neural network after a single forward and backward pass. For instance, (Han et al., 2015) and other researchers have employed static saliency measurements such as weight magnitude as a criterion, while (Lee et al., 2022) have utilized backward information as a saliency measurement, such as gradient values. Furthermore, other studies (Lee et al., 2018; Wang et al., 2019; Abdelfattah et al., 2021) have enhanced this approach by extracting information from a single forward and backward pass to compute predefined saliency metrics, subsequently used to gauge the significance of parameters within the neural network.

While these prior works may have explored various aspects, such as layer-wise or parameter-wise granularity, our research focuses exclusively on a layer-wise perspective. Our present study builds upon these previous endeavors to identify crucial parameters. We employ analogous saliency metrics and combine them to rank each layer, followed by fine-tuning. This ensemble approach enables our method to leverage the relationships and insights among these metrics, resulting in a more consistent and accurate ranking of the network's layers.

## 3 METHOD

Consider a pre-trained model $\mathcal{F}_\Theta(\cdot)$ with a set of parameters $\Theta = \{\mathbf{W}_1, \mathbf{W}_2, \ldots, \mathbf{W}_n\}$. Here, $W_i$ refers to the parameters in the $i$-th layer among a total of $n$ layers. This pre-trained model can be adapted to downstream tasks and each downstream task is represented by a training dataset of context-target pairs, denoted by $\mathcal{Z} = \{(x_i, y_i)\}_{i=1,..,N}$, where both $x_i$ and $y_i$ are training data and label. In vision tasks, $x_i$ represents the image and $y_i$ represents its corresponding label. In language tasks, $x_i$ represents the input tokens and $y_i$ represents the reference results.

**Vanilla Full Fine-Tuning** During full fine-tuning, the model is initialized with pre-trained weights $\Theta_0$ and updated to $\Theta_0 + \Delta_\Theta$ by repeatedly following the gradient to optimize the following objective.

$$\min_\Theta \sum_{(x,y) \in \mathcal{Z}} \mathbb{L}\left(y, F_\Theta(x, y)\right) \tag{1}$$

Where $\Delta\Theta = \nabla \mathcal{F}_\Theta(\mathbf{X})$, and $\mathbf{X} \in \mathcal{Z}$ and $\mathcal{Z}$ is the training dataset. Vanilla fine-tuning has the greatest number of learnable parameters $\|\Delta_\Theta\| = \|\Theta\|$, as well as the largest training cost

**Sparse Fine-Tuning.** For sparse fine-tuning, only a smaller number of parameters is trained and the set of learnable parameters is frozen during the fine-tuning process.

$$\min_{\Phi(\Theta)} \sum_{(x,y)\in\mathcal{Z}} \mathbb{L}\left(y, F_{\Theta_0}(x,y)\right) \tag{2}$$

where $\Phi(\Theta)$ is a sparse subset of $\Theta$ and $\|\Delta_{\Phi(\Theta)}\| << \|\Theta\|$. Studies (Zaken et al., 2021; Cai et al., 2020; Lee et al., 2022) have shown that such a paradigm can achieve similar and even sometimes outperform full fine-tuning. Our ZEST aims to highlight the value of carefully choosing the subset of learnable parameters and proposes a solution to efficiently find such a scheme.

## 3.1 Zero-shot Metrics and Contribution Scores

Many studies have attempted to assess the significance of layers for a particular fine-tuning task and have suggested that not all layers contribute equally to the fine-tuning process. Based on this observation, an intuitive idea is that this "importance" can be correlated with certain model metrics. In this work, we concentrate on Zero-shot Metrics, which are easier to obtain and can be computed at a low cost. Specifically, we evaluate metrics within the following three categories.

**Zero-Shot Static Metrics.** Before executing the model, the weight itself already contains plenty of information. Previous explorations have shown that pruning weights with smaller magnitudes can reduce storage and even improve generalization (Han et al., 2015), while removing weights with larger variance can significantly impair model performance (Bau et al., 2020). Therefore, in this study, we include the mean and variance of weights and analyze their relationship with fine-tuning accuracy.

$$\texttt{Wt.Avg} = ||\Theta||, \quad \texttt{Wt.Std} = \hat{\Theta}$$

**Zero-Shot Forward Metrics.** The forward pass is directly associated with the model accuracy. The activation values of each layer have a numerical impact on the final prediction. Following Lee et al. (2022), we also include the activations of each layer as a metric and study their connection with the final fine-tuning performance.

$$\texttt{Act.Avg} = \bar{X}, \quad \texttt{Act.Std} = \hat{X}, \quad |\texttt{Act}|.\texttt{Avg} = |\bar{X}|, \quad |\texttt{Act}|.\texttt{Std} = |\hat{X}|$$

**Zero-Shot Backward Metrics.** During the fine-tuning process, we not only perform the forward pass but also the backward pass to compute gradients and update parameters. When working with a new dataset that changes rapidly, it is reasonable to assume that a layer with large gradient magnitudes in each iteration is more influential for accuracy. Therefore, we include several metrics related to the backward pass. The first two metrics are the mean and variance of gradients. Additionally, we use the dot product of the weight and its corresponding gradient as a simple estimation, referred to as "plain."

$$\texttt{Grad.Avg} = \bar{G}, \quad \texttt{Grad.Std} = \hat{G}, \quad |\texttt{Grad.}|\texttt{Avg.} = |\bar{G}|, \quad |\texttt{Grad.}|\texttt{Std} = |\hat{G}|, \quad \texttt{plain} = \overline{G \cdot \Theta}$$

Besides simple statistical metrics, we further include more measurements from recent work:

- `snip`: Lee et al. (2018) estimates the saliency of each layer in the network by evaluating the importance of weights based on their sensitivity to the loss with respect to network inputs.
- `grasp`: Wang et al. (2020) estimates the importance of each layer based on the sensitivity to the |grad| (as opposed to loss in `snip`) with respect to the network inputs.
- `synflow`: Tanaka et al. (2020) introduce this refined variation of the concept of synaptic saliency scores. Unlike methods like snip or grasp, which rely on a minibatch of training data and cross-entropy loss, the synflow technique calculates a loss function derived straight-forwardly from the product of all the network parameters. Consequently, there is no need for training data to compute this loss or the synflow metric itself.
- `fisher`: Theis et al. (2018) measure channel saliency by removing activation channels (and their corresponding parameters) that are estimated to have the least effect on the loss. The metric is computed by taking the norm of the activation multiplied by the gradient.

$$\texttt{snip} = |\frac{\partial\mathcal{L}}{\partial\Theta} \odot \Theta|, \quad \texttt{grasp} = -(H\frac{\partial\mathcal{L}}{\partial\Theta}) \odot \Theta, \quad \texttt{synflow} = \frac{\partial\mathcal{L}}{\partial\Theta} \odot \Theta, \quad \texttt{fisher} = (X \odot G)^2$$

$\mathcal{L}$ is the corresponding loss function of the network of the neural network, $\Theta$ is the weights, $G$ is the gradient, $X$ is the activation, and $\odot$ is the Hadamard product.

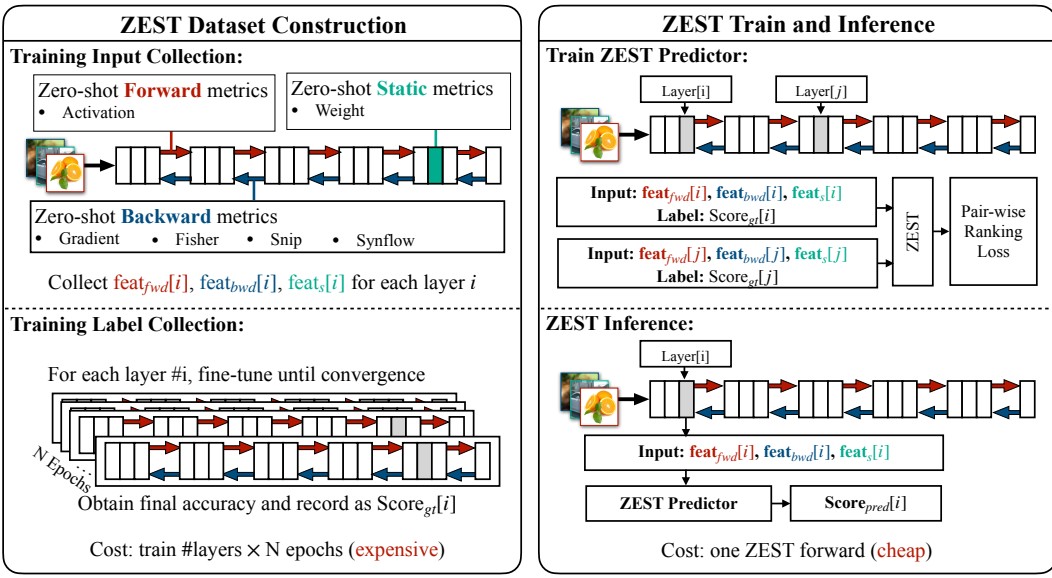

**Figure 2.** The overview of our proposed ZEST. *Left*: We first collect zero-shot forward, backward, and static metrics from the model. Then, we iterate each layer and record the one-layer fine-tuning accuracy as the ground truth score. *Right*: During method training, each iteration, we sample two layers and feed the pre-collected zero-shot metrics into the predictor. After finishing training, when a new dataset comes in, ZEST only requires one single minibatch and then can accurately estimate the layers' importance based on their score.

**Contribution Score.** We next seek metrics to quantitatively evaluate the zero-shot metric performance. Inspired by Lin et al. (2022); Lee et al. (2022), we adapt the one layer/block fine-tuning accuracy as the contribution score. For each layer, we freeze all other parameters except the last classification layer and fine-tune them until convergence. The final fine-tuning performance reveals how much each layer contributes to the accuracy of the specific downstream task.

The contribution score has shown to be able to find the most important top-k layers to speed up fine-tuning Lin et al. (2022), but this method can be costly and unscalable for modern models. Taking LLaMA as an example, the model contains 32 blocks where each block contains 7 learnable layers, excluding the layer norm. The 7 learnable layers are as follows, the 3 $q$, $k$, and $v$ linear layers of the attention module, and the 4 $out$, $gate$, $up$, and $down$ projection linear layers of the MLP ($q\_proj$, $k\_proj$, $v\_proj$, $o\_proj$, $gate\_proj$, $up\_proj$, $down\_proj$). It results in a total of 224 layers which means we need to fine-tune LLaMA 224 times to obtain the full contribution score on one dataset. With 7 different datasets, we fine-tuned LLaMA for a total of 1568 times.

### 3.2 ZEST: ZEROSHOT SPARSE FINE-TUNING

Recognizing that a single, static set of zero-shot metrics may not provide a comprehensive assessment of layer contribution within a neural network(gone into finer detail in section 4.2), we introduce ZEST, which leverages all available metrics to automatically identify significant layers. Shown in Figure. 2, the process consists of three distinct phases: (i) the training phase of the ZEST predictor, (ii) the inference phase of the ZEST predictor, and (iii) the sparse fine-tuning phase. These phases collectively allow us to harness the full range of metrics and effectively pinpoint the layers that carry the most significance in the context of the contribution to the whole neural network's performance.

**Training of the ZEST Predictor** We initiate the process by gathering expensive layer-wise contribution analyses for all possible combinations of datasets and models, establishing this as our reference or ground truth. Once the dataset collection is complete, we perform inference on each permutation, utilizing only a single mini-batch, in order to derive the aforementioned cheap zero-shot metrics.

Subsequently, with the zero-shot metrics serving as our input and the contribution analysis serving as the reference or ground truth, we proceed to train a predictor. During the training phase, each pair of

examples originates from the same model-dataset combination. However, it's important to note that pairs within a batch may come from various model-dataset combinations.

Furthermore, it's worth mentioning that the pairs sourced from the testing model-dataset remain uncontaminated throughout this process. This setup creates a transferable scenario in which the predictor learns to generalize across different model-dataset combinations.

**Inference of ZEST Predictor**  When dealing with new and previously unseen test model-dataset pairs, our approach involves conducting a single forward and backward pass using a single mini-batch. This process allows us to obtain cost-effective zero-shot metrics for each layer within the model. Once we've acquired these inexpensive metrics, we employ our ZEST predictor for each layer separately. This predictor provides us with the corresponding ZEST score for each layer.

**Sparse Fine-Tuning**  Subsequently, we proceed to rank each layer within the given model-dataset combination based on the ZEST score it receives. In the fine-tuning phase that follows, we select the top $n$ layers that have the highest ZEST scores. These selected layers are considered the most important ones for fine-tuning, as they are expected to have the greatest impact on the overall performance of the model in the context of the specific model-dataset combination.

## 4 EXPERIMENTS

In this section, we start by presenting the details of our experimental setup and then evaluate the reliability of individual zero-shot metrics. We assess the effectiveness of ZEST and analyze its performance in real-world fine-tuning scenarios, providing a comprehensive evaluation of its capabilities.

### 4.1 SETUP

To assess the effectiveness of the ZEST, we conduct comprehensive experiments encompassing both vision and language tasks. To maintain fairness in our evaluation, we take care to ensure that the datasets employed in training the ZEST predictor are distinct from those used for accuracy assessment. This separation of data sources ensures an unbiased evaluation of the ZEST's performance.

For vision tasks, we choose three widely-adapted architectures: MobilenetV2 (Sandler et al., 2018), EfficientNet (Tan & Le, 2019), and ResNet-50 (He et al., 2016). We evaluate these architectures on eight different downstream transfer learning datasets: Aircraft, Cars, CIFAR10, CIFAR100, CUB , Flowers, Food, and Pets (Bossard et al., 2014; Maji et al., 2013; Krause et al., 2013; Krizhevsky et al., 2009; Wah et al., 2011; Nilsback & Zisserman, 2008; Parkhi et al., 2012). We use {CIFAR-10, Aircraft, Cars, Flowers, Food, and Pets} datasets to construct the ZEST predictor and evaluate the effectiveness of the remaining CIFAR-100 and CUB. We use a learning rate of $3e^{-4}$ and $6e^{-4}$ for CIFAR100 and CUB respectively with decay $5e^{-4}$. The model is trained for 30 epochs.

In our evaluation of natural language tasks, we utilize GLUE dataset  (Wang et al., 2018) for BERT (Devlin et al., 2019). For LLaMA (Touvron et al., 2023), we choose the Arc-C, Arc-E, Hellaswag, OpenbookQA, PIQA, and Sciq (Bisk et al., 2020; Clark et al., 2018; Mihaylov et al., 2018; Welbl et al., 2017; Zellers et al., 2019) datasets. We report the performance of BERT on SST2 and MRPC, and the performance of LLaMA on Arc-E and HellaSwag. The remaining datasets within GLUE and LLaMA are used for training the ZEST predictor. These results are obtained from training for three epochs. The learning rate is set to $2e^{-5}$ for LLaMA and $5e^{-5}$ for BERT. Additionally, a weight decay of $1e^{-2}$ is applied during training. This configuration ensures a consistent evaluation and comparison across the various datasets and models in our natural language tasks.

We conduct all experiments on Nvidia A100 SXM 80G GPUs with PyTorch 2.0, Torchvision 0.15. For language-related experiments, we use Huggingface Transformers 4.30 and PEFT v0.4.0. We will release our codebase when less anonymous.

### 4.2 ROBUSTNESS OF SINGLE ZERO-SHOT METRICS

It is important to evaluate the reliability of zero-shot metrics. Our goal is to analyze the relationship between zero-shot metrics and contribution scores, which are considered as the ground truth. In Table. 1, we compare the performance and report the Kendall Tau correlation (Kendall, 1938), which

**Table 1.** The zero-shot metric scores of MobilenetV2 (Sandler et al., 2018), EfficientNet (Tan & Le, 2019) and ResNet-50 (He et al., 2016). We report the Kendall Tau(↑) correlation between each metric and the ground truth where 'cf100' and 'cub' are the abbreviations of CIFAR-100 and CUB200 datasets. The best-performing 3 zero-shot metrics are highlighted in red, blue, and green.

| Model | | LLAMA | | BERT | | MblnetV2 | | EfficientNet | | ResNet50 | |
| Dataset | | arc-e | hellaswag | sst2 | mrpc | cf100 | cub | cf100 | cub | cf100 | cub |
|---|---|---|---|---|---|---|---|---|---|---|---|
| Backward | \|Grad\| | 0.04 | 0.17 | 0.64 | 0.68 | -0.42 | 0.84 | 0.66 | 0.83 | -0.32 | 0.66 |
| | Snip | 0.37 | 0.23 | 0.49 | 0.41 | -0.41 | -0.38 | -0.21 | -0.21 | -0.49 | -0.46 |
| | Grasp | -0.02 | 0.05 | 0.20 | -0.03 | -0.30 | -0.39 | -0.27 | 0.04 | -0.49 | -0.38 |
| | Fisher | -0.15 | 0.04 | -0.26 | 0.20 | -0.23 | -0.12 | -0.14 | -0.07 | -0.44 | -0.22 |
| | Plain | 0.13 | -0.02 | -0.17 | 0.02 | 0.21 | 0.07 | -0.23 | 0.04 | -0.08 | -0.18 |
| | Synflow | -0.04 | -0.06 | -0.46 | -0.01 | -0.41 | -0.24 | -0.16 | -0.10 | -0.08 | -0.32 |
| Forward | Avg. Act | -0.04 | -0.05 | -0.38 | -0.27 | -0.15 | -0.09 | -0.14 | -0.07 | 0.21 | -0.06 |
| | Std. Act | -0.02 | -0.07 | -0.13 | 0.18 | 0.33 | 0.06 | 0.25 | 0.07 | -0.38 | -0.38 |
| | Avg. \|Act\| | -0.02 | -0.07 | 0.07 | 0.41 | 0.27 | -0.01 | 0.20 | 0.03 | -0.38 | -0.35 |
| | Std. \|Act\| | -0.04 | -0.05 | 0.05 | -0.03 | 0.38 | 0.12 | 0.19 | 0.03 | -0.37 | -0.36 |
| Static | Avg. Wt | -0.21 | -0.49 | -0.38 | -0.30 | -0.28 | -0.19 | 0.09 | 0.11 | 0.05 | 0.19 |
| | Std. Wt | -0.24 | -0.50 | -0.25 | -0.05 | -0.57 | -0.51 | -0.30 | -0.34 | -0.43 | -0.55 |
| | Avg. Grad | -0.11 | -0.06 | 0.05 | -0.07 | 0.22 | 0.09 | 0.09 | -0.06 | 0.31 | 0.12 |
| | Std. Grad | -0.04 | -0.05 | 0.51 | 0.09 | -0.47 | -0.51 | -0.21 | -0.25 | -0.49 | -0.58 |

**Table 2.** The comparison of Kendall Tau correlation of the **best** zero-shot metric, and ZEST, where 'Cf100' and 'StfC' denote for CIFAR-100 and Stanford-Cars datasets, respectively. 'Best ZS' indicates the best zero-shot metric for each dataset and 'ZEST' denotes the single and universal predictor for all datasets. The higher the Kendall Tau is, the more accurate the predicted ranking is. We find that our trained ZEST predictor significantly outperforms the single zero-shot metrics.

| | LLaMA | | BERT | | MobilenetV2 | | EfficientNet | | ResNet50 | |
| | arc-c | hellaswag | sst2 | mrpc | cf100 | cub | cf100 | cub | cf100 | cub |
|---|---|---|---|---|---|---|---|---|---|---|
| Best ZS | 0.37 | 0.23 | 0.64 | 0.68 | 0.38 | 0.68 | 0.66 | 0.40 | 0.31 | 0.25 |
| ZEST | **0.74** | **0.82** | **0.70** | **0.83** | **0.88** | **0.81** | **0.78** | **0.78** | **0.71** | **0.82** |

measures the similarity of the ordering. For each column, we first train each layer individually until convergence and obtain the contribution score as the ground truth. Then, we compute the Kendall Tau correlation between two orders: one ranked by contribution score and another ranked by zero-shot metrics. We study the correlation between them. Our observations are as follows:

**Different Models Prefers Different Zero-shot Metrics.** As demonstrated in Table 1 for CIFAR-100 results, we can see that smaller models like MobilenetV2 and EfficientNet tend to favor backward zero-shot metrics. However, larger models like ResNet-50 indicate that static zero-shot metrics offer a more accurate representation of layer importance.

**Different Datasets Prefers Different Zero-shot Metrics.** When examining two sub-columns for each dataset, we highlight the top three related metrics using the colors red, blue, and green. We discovered that even for the same model and weights, the best related zero-shot metrics can vary depending on different input distributions. For instance, MobilenetV2 favors Zero-Shot Forward Metrics for CIFAR-100 because this dataset is similar to its pre-training set (ImageNet). However, for downstream tasks with different distributions (CUB), the Zero-shot Backward metrics exhibit a much higher correlation, highlighting the importance of updating weights.

Zero-shot metrics demonstrate a strong correlation with contribution scores, indicating that these metrics inherently contain information about a layer's importance. However, the best metrics can vary significantly for different models and downstream tasks. Therefore, we propose using ZEST to combine them.

## 4.3 ZEST Accurately Estimates the Layers' Importance

Our model accurately estimates the importance of each layer during fine-tuning. Table. 2 presents the Kendall Tau (higher is better) of ZEST's predictions and the best zero-shot metrics for each

{dataset, model} combination. By analyzing various zero-shot metrics, ZEST effectively estimates the importance and significantly outperforms previous single zero-shot metric results.

Furthermore, the ZEST assessment consistently demonstrates a strong correlation with ground truth contribution scores across different models and datasets, indicating its general and universal capability. By identifying the layers that have a significant impact on fine-tuning accuracy, models can prioritize optimizing those specific layers when encountering a new dataset. This leads to improved model performance (Section. 4.4) and enhanced training throughput (Section. 4.6).

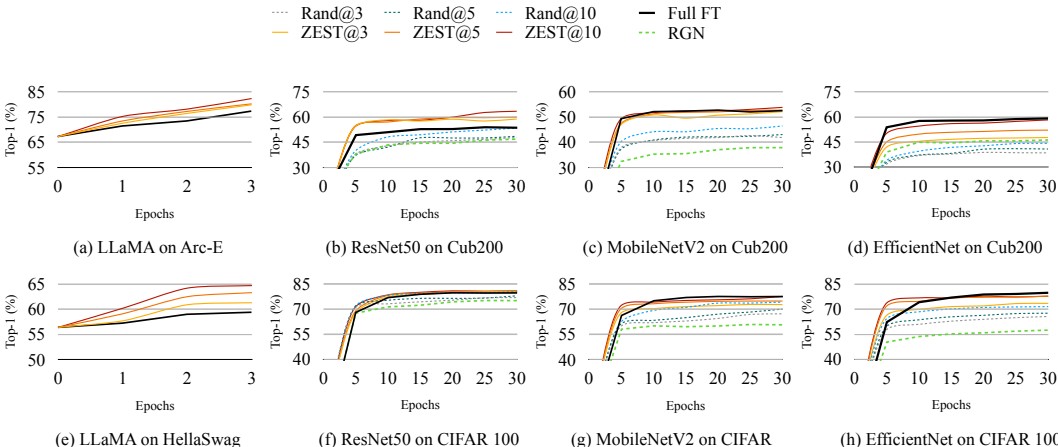

**Figure 3.** The end-to-end validation curve of our ZEST. Each training curve is averaged from 3 runs, and we denote the number of selected layers in the legend. We observe that ZST results closely match the Full-FT results (black line) and consistently outperform the random baselines (dashed lines).

## 4.4 ZEST End-to-End Performance Comparison

We present the end-to-end fine-tuning results via only updating layers chosen by ZEST. To show the generalization and universal ability of our ZEST, we conduct experiments on both vision datasets (MobilenetV2, EfficientNet, ResNet-50) and language tasks (BERT and Llama).

For vision models, we plot the validation curve of fine-tuning in Figure. 3. We thoughtfully compare the cases when picking {3, 5, 10} layers from the model and fine-tune until converge. Suprisely, we find that learning 5 to 10 layers can perform on par or outperform full fine-tuning (FT-Full), suggesting the vast redundancy in conventional fine-tuning baselines. Further, results yield by ZEST not only shows higher performance, but also faster convergence than random baselines. This provides a strong evidence of the effectiveness of ZEST.

For language models such as BERT and LLaMA, updating 3 to 5 layers of BERT (84 layers) or Llama (224 layers) can closely match, or even outperform full fine-tuning results (Table. 3), where 3 layers are only 0.013% of the whole model parameters. Further, we notice when compared with fine-tuning using ground-truth contribution scores (expensive but most accurate), ZEST chosen layers consistently match the performance and offer valuable guidance in dissecting and finding each layer's contribution to the overall prediction. In contrast, previous methods such as RGN fails to capture the most important layers.

## 4.5 ZEST Comparison & Combination with PEFT

It is worth highlighting that our ZEST approach is orthogonal to many Parameter-Efficient Fine-tuning (PEFT) methods. This means that ZEST can be seamlessly combined with PEFT methods like LoRA, resulting in a hybrid update scheme. In this hybrid approach, ZEST first identifies layers with higher importance values for fine-tuning, and then LoRA is employed to introduce low-rank adaptation branches that facilitate the fine-tuning process.

In the experimental results presented in the last section, specifically in Table 3, you can observe the performance of this combined approach. In this table, "LoRA (rank=8)" can be considered as "LoRA + ZEST@all layers". It is worth noting that When comparing models with similar learnable

**Table 3.** Performance comparison of LLAMA-7B (Touvron et al., 2023) on QA, and BERT (Devlin et al., 2018) on GLUE benchmarks. The best and second-best results are highlighted with color. We can see that ZEST achieves comparable performance with full fine-tuning, and ZEST @10 performs consistently better than full fine-tuning. Since ZEST is orthogonal to LoRA, we also do a combination of the two.

| Model | LLaMA | | | BERT | | |
|---|---|---|---|---|---|---|
| Dataset | arc-e | hellaswag | #Params | sst2 | mrpc | #Params |
| Zero-shot | 67.3 | 56.4 | \ | 50.9 | 70.6 | \ |
| FT-Full | 77.4 | 59.4 | 7B | 91.1 | 84.5 | 86M |
| Contribution@3 | 80.0 | 61.3 | 90M | 91.0 | 78.9 | 1.2M |
| Contribution@5 | 82.0 | 63.8 | 165M | 91.4 | 82.7 | 2.4M |
| ZEST@3 | 79.9 | 61.3 | 90M | 90.9 | 78.9 | 1.2M |
| ZEST@5 | 82.4 | 63.5 | 165M | 91.4 | 82.7 | 2.4M |
| ZEST@10 | 82.8 | 64.2 | 338M | 91.6 | 85.0 | 4.8M |
| LoRA (rank=8) | 74.5 | 59.8 | 8M | 91.3 | 81.4 | 0.3M |
| LoRA+ZEST@3 | 74.3 | 59.9 | 0.8M | 85.6 | 76.3 | 0.018M |
| LoRA+ZEST@5 | 74.6 | 59.9 | 1.3M | 87.7 | 77.3 | 0.03M |
| LoRA+ZEST@10 | 74.7 | 60.1 | 2.6M | 89.9 | 82.3 | 0.06M |

parameters, such as "ZEST@3" and "LoRA (rank=8)", it is evident that ZEST, when applied to selected layers, outperforms the universal LoRA results. Additionally, the combination of ZEST and LoRA achieves fine-tuning performance that closely matches that of LoRA, despite having fewer learnable parameters. This demonstrates the effectiveness of ZEST in guiding the fine-tuning process and optimizing the use of parameters. These results highlight the independence of ZEST and its ability to enhance the performance of fine-tuning methods like LoRA while reducing the required number of parameters.

**Table 4.** The efficiency comparison of ZEST and other baselines. For vision models, we set the input resolution to 224 and the batch size to 32. For language models, we set the batch size to 1 and the sequence length to 512. ZEST effectively finds valuable layers, leading to the loss of fine-tuning memory and latency as well as the comparable final model quality.

| | LLaMA | | BERT | | MobileNetV2 | | ResNet50 | |
|---|---|---|---|---|---|---|---|---|
| | lat | mem | lat | mem | lat | mem | lat | mem |
| Full FT | 179ms | 79G | 25.2ms | 3.8G | 18.2ms | 3.9G | 27.3ms | 4.3G |
| ZEST @5 | 75ms | 29G | 17.8ms | 2.9G | 9.2ms | 1.9G | 16.9ms | 2.1G |

## 4.6 ZEST END-TO-END EFFICIENCY COMPARISON

One advantage of accurate estimation of important layers is the fine-tuning efficiency improvement. By updating fewer layers during fine-tuning, the model no longer needs to back-propagate to the very first layers. Therefore, with ZEST chosen layers, we observe training cost reductions. Specifically in Table 4. We can see that ZEST can efficiently decrease the latency and the memory consumption during the training phase. In particular. For vision models, the memory consumption and latency are decreased by roughly 50% for ResNet50, and 40% for MobileNetV2. For language models, the latency and memory are both roughly decreased by 40% for BERT, and 60% for LLaMA.

## 5 CONCLUSION

In summary, we have introduced a novel method, referred to as ZEST, for the rapid identification of important layers within a neural network. Importantly, ZEST operates without altering the model's architecture or input data, making it applicable across various tasks and modalities. By employing ZEST, we leverage the advantages of sparse fine-tuning, leading to improved task performance compared to full fine-tuning, while concurrently achieving reductions in memory usage and latency by up to 2.4x and 2.7x, respectively.

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

## A  APPENDIX

You may include other additional sections here.

