# OpenReview forum: "ZEST: ZEROSHOT SPARSE FINE-TUNING"
_ICLR.cc/2024/Conference — Submitted to ICLR 2024_

### Official Review · Reviewer_cd1r · 2023-10-30

**Soundness:** 2 fair
**Presentation:** 2 fair
**Contribution:** 2 fair
**Rating:** 5
**Confidence:** 4

**Summary:**

Previous studies have found that fine-tuning a small subset of layers in a model can achieve comparable or even better performance than full fine-tuning. These methods were based on metrics, known as zero-shot metrics, that measure the importance of network layers to select the optimal one for fine-tuning. However, this paper points out that these metrics perform inconsistently across different models and datasets, making it challenging to use a single metric for universal layer selection. To address this, the paper establishes a dataset that consists of various metrics and their corresponding optimal layers and trains a universal predictor for optimal layer selection using this dataset. Experimental results demonstrate that the predictor generalizes well across different models and datasets, improving fine-tuning performance while enhancing efficiency.

**Strengths:**

The problem addressed is interesting and relevant, especially considering the increasing model's size in the current era.

**Weaknesses:**

1. The paper does not provide a detailed configuration of the optimal fine-tuning layers for different models on different datasets, making it difficult for readers to identify patterns. Additionally, there is confusion regarding the claim that fine-tuning only a subset of layers can accelerate the process. Generally, fine-tuning deeper layers can be faster since backward propagation does not reach shallower layers. However, achieving downstream generalization often requires fine-tuning parameters in the shallower layers, making it challenging to achieve significant acceleration. Therefore, the authors should provide a specific layer selection for each case and provide a clear explanation of how acceleration is achieved.
2. The experiments in the paper are not extensive. Since the training is conducted on related models and datasets, there is a high risk of overfitting. Therefore, the proposed method should be validated on a wider range of models and datasets. For example, for visual tasks, validation should be performed on models like ViT, Swin, and VTAB-1K benchmark. For NLP tasks, validation should be performed on newer models like Kosmos, LLaMA 2, and LLMs. Additionally, further validation should be conducted on current multimodal large models such as BLIP-2, MiniGPT-4, LLaVA, etc. Furthermore, the architecture of the ZEST predictor is not clearly explained.
3. The paper indicates that ZEST is orthogonal to many PEFT methods but only validates its compatibility with LORA in NLP tasks. Considering the existence of many new state-of-the-art PEFT methods, can ZEST also be combined with these methods? For example, in visual tasks: Adaptformer, ConvPass, Fact, and in NLP tasks: Adapter, Prompt tuning, Ladder-side tuning, etc.
4. Table 1 indicates that a higher Kendall Tau value is better. However, in fact, negative values indicate a negative correlation between variables. Shouldn't the absolute value be used to measure the degree of the correlation?

**Questions:**

No other questions

---

> ### Author Response · Authors · 2023-11-20
>
> ### **Q1: Configuration of the optimal fine-tuning layers**
>
> We thank the reviewer for raising this, and we would like to point out that **different models/datasets have different preferences for different layers**, and there is no single metric that can choose the best layers. Thus we propose ZEST, which can choose layers dynamically. As a simple proof, the authors list the top-10 indices of preferred layers of Resnet50 on Cifar100 and Cub200.
>
> Resnet50 on Cifar100: 41, 25, 43, 42, 39, 40, 37, 38, 35, 31, 36
>
> Resnet50 on Cub200:  23, 13, 25, 24, 12, 11, 28, 41, 14,  27,1 7
>
> LLaMA on OpenbookQA: 73, 87, 115, 63, 67, 94, 95, 101, 108, 104
>
> LLaMA on Arc-Easy: 76, 75, 67, 70, 77, 68, 62, 74, 69, 49
>
> We observe that while they have a small amount of overlap, they are mainly uncorrelated. We also observe that the shallowest top-10~25 best layers to finetune often appear midway of the whole network depth, thus achieving an efficient speedup.
>
> ### **Q2: ZEST Evaluations with Backbone Models**
>
> We thank the reviewer for going into such detail, we have also made additional experiments, but due to the short rebuttal window, we only implemented **mobileViT** and **LLaMA2**.
>
> |  |MobileViT |  | LLaMA2 |  |  |
> | --- | --- | --- | --- | --- | --- |
> |  | Cifar100 | Cub200 | Arc-C | Arc-E | PIQA |
> | FFT | 78.9 | 56.9 | 51.1 | 80.8 | 79.5 |
> | ZEST @ 3 | 71.2 | 53.9 | 50.9 | 80.5 | 79.3 |
> | ZEST @ 5 | 75.8 | 56.8 | 51.4 | 80.5 | 79.4 |
> | ZEST @ 10 | 78.5 | 57.5 | 52.5 | 82.2 | 80.0 |
>
> As shown in the Table, ZEST continues to properly find important layers for each downstream task and demonstrates that fine-tuning 3~5 layers could achieve on-par accuracy with full fine-tuning. We will include them in the next revision
>
> ### **Q3: ZEST Evaluations with More PEFT Methods**
>
> Due to the limited time window, we were only able to implement the adapters on BERT. Below is a table showing our results of combining the adapter and ZEST.
>
> Besides more backbones, we also experimented with more PEFT methods to show the general ability of ZEST and attach the results of ZEST + Adapter below. The workflow is that Adapter first inserts bottleneck layers and use ZEST to find important layers between them and then fine-tune.
>
> Adapter results for BERT:
>
> | BERT | SST2 | MRPC |
> | --- | --- | --- |
> | FT-Full | 91.1 | 84.5 |
> | Grad_Norm @ 3 | 87.5 | 76.6 |
> | Grad_Norm @ 5 | 88.6 | 77.9 |
> | Grad_Norm @ 10 | 89.8 | 80.1 |
> | ZEST @ 3 | 90.9 | 78.9 |
> | ZEST @ 5 | 91.4 | 82.7 |
> | ZEST @ 10 | 91.6 | 85.0 |
> | Adapter | 90.5 | 82.8 |
> | Adapter + ZEST @ 3 | 89.6 | 81.2 |
> | Adapter + ZEST @ 5 | 90.1 | 82.6 |
> | Adapter + ZEST @ 10 | 91.0 | 83.2 |
>
> ZEST consistently demonstrates higher performance than Adapter. We will follow the reviewer's advice to experiment with more PEFT methods. But this time, due to the limited rebuttal window, we can only provide the Adapter results on BERT.
>
> We hope that our responses have addressed your concerns. Please let us know if there are any other experiments or clarifications we can provide to convince you to increase the rating.

---

> > ### Author Response · Authors · 2023-11-22
> >
> > Dear reviewer,
> >
> > Thank you once again for providing high-quality reviews. As today is the final day for discussion, could you please let us know if our response has fully answered your questions? We would be more than happy to provide further details and responses for any additional questions you may have.
> >
> > Best regards,
> >
> > Authors

---

### Official Review · Reviewer_KjeW · 2023-11-01

**Soundness:** 4 excellent
**Presentation:** 4 excellent
**Contribution:** 3 good
**Rating:** 6
**Confidence:** 2

**Summary:**

Traditionally, problems that go about determining the importance of different layers/neurons such as in pruning, or masked/sparse fine-tuning etc., use metrics that are derived from the network's performance on the target dataset for which the alterations to the model are performed. Modern approaches, learn a meta-model are a function of the model weights, input dataset and other factors and produce a rating for each layer/neuron. The authors propose one such meta-model for identifying which layers are to be frozen and which made adaptable for fine-tuning purposes, in order to optimize efficiency. They learn this meta-model using an array of previously collected "training data" of model-dataset-metric combinations. During inference time, their work does not need to access the entire target dataset, but only one sample from it, hence "zero-shot".

**Strengths:**

The concept of associating a metric with a layer of a neural network for either compression, such as pruning etc., has been a searching one. Ideas ranging from using forward metrics, such as activations, backward metrics such as gradients and their higher-order approximations such as Fisher co-efficients have been tried and shown some success. Several works also successfully demonstrate that in larger networks, even randomly associated neuron/layer importance/saliency can be a good proxy, due to the incredible plasticity of neural networks to learn to new data.

In the case of fine-tuning, however, plasticity can become a trouble for generalization, due to catastrophic forgetting. This is particularly so in cases where the fine-tuning dataset target is a small and hard dataset, which often leads to overfitting. Therefore, in cases of sparse fine-tuning, the choice of which parameter to freeze and which not to, is important. The authors suppose that using metrics that measure importance might not be the ideal way and instead learn a meta-model that can predict a layer-wise importance measure. They then use the top-n layers from this measure for fine-tuning. Although not completely original, this is novel and is sound reasoning.

The standard metrics used to support their cases are also used to learn their meta-model and these are well-chosen. The experiments show that there is some value in training a meta-model, by being better than almost all standard metrics. Considering that their method abstracts out the metric measurement model from the system for the actual sparse fine-tuning, their work can be seamlessly combined with other fine-tuning techniques.

**Weaknesses:**

The work has several weaknesses that need to be discussed/addressed.
1. While it is clear from table 2, that the meta-model estimation is better than the best standard metrics and therefore can generalize for all model/dataset combinations and from figure 3 is shown to be better than random, it is not clear how much better this model is when compared against the standard metrics. This is important to know since the meta-model itself requires training and the data-collection and training is significantly expensive.
2. Table 3 is a good result. It shows comparisons against another fine-tuning technique. The reviewer would encourage the authors to add their "static predictors" as well to this table.

**Questions:**

The reviewer has one question, which requires further results. How much benefit is this predictor yielding when compared against the other static predictors that it is learning from? This needs to be weighed against, how expensive it is to collect data for training this predictor model and training it itself. Also, how does that delta change with model-size/type and target dataset-size/type.

---

> ### Author Response · Authors · 2023-11-20
>
> We thanks reviewer #KjeW appreciates ZEST’s contribution and novelty. We perform and report additional ablation experiments to better answer your question
>
> ### **Q1: Comparison between ZEST and Static Predictors**
>
> We conducted experiments to compare ZEST with static predictors ( GradNorm is chosen because of the high relevance) and attach results below. Do note that although GradNorm may have high Kendall’s Tau correlation (indicating it’s general ability to rank every layer), it’s top-n accuracy is not high (indicating it’s inability to choose the best layers).
>
> |  | Resnet50 |  | LLaMA |  | BERT |  |
> | --- | --- | --- | --- | --- | --- | --- |
> |  | Cifar100 | Cub200 | Arc-e | Hellaswag | MRPC | SST2 |
> | Grad_Norm @ 3 | 77.0 | 53.9 | 77.5 | 58.5 | 76.6 | 87.5 |
> | Grad_Norm @ 5 | 80.0 | 57.1 | 78.4 | 59.0 | 77.9 | 88.6 |
> | Grad_Norm @ 10 | 80.6 | 58.1 | 79.1 | 59.2 | 80.1 | 89.8 |
> | ZEST @ 3 | 80.3 | 58.9 | 79.9 | 61.4 | 78.9 | 90.9 |
> | ZEST @ 5 | 80.7 | 60.0 | 82.4 | 63.5 | 82.7 | 91.4 |
> | ZEST @ 10 | 80.9 | 63.5 | 82.8 | 64.2 | 85.0 | 91.6 |
>
> For small vision models, the delta change between static predictor and ZEST are more significant when updating fewer layers. For CIFAR100,  when updating only three layers, GradNorm demonstrates accuracy of 77.0% while ZEST achieves 80.3%  (2.3% higher).
>
> For medium and large language models, the improvement of ZEST becomes much more significant compared to static predictors: From the table, ZEST consistently outperforms static predictor by large margin, and for BERT, the accuracy of fine-tuning 3 layers (found ZEST) is even higher than fine-tuning 10 layers (using static predictor).
>
> These results show that ZEST can accurately choose the “important” layers for fine-tuning, achieving higher accuracy with lower fine-tuning cost. We will perform ablation studies on more static predictors and include them in the next revision.
>
> ### **Q2: The Cost of Training the ZEST Predictor and Each Model?**
>
> This is indeed a good and related question! For the ZEST predictor, it takes 150 GPU hours and 2040 GPU hours for vision (ResNet-50) and language tasks (Llama-7b) respectively. For single model training, ResNets and Llama-7B takes 0.5 and 1.5 hours respectively. However, we would like to emphasize that **such a predictor pre-training only needs to be performed once** and can be generally applied to various downstream tasks.  Considering building 10,000 customized chatbots using datasets similar as Stanford Alpaca[1]
>
> **Cost without Zest**
>
> 0.179s (single iteration latency) * 52,000 (dataset size) * 3 (epochs) * 10,000 (#num of tasks) = 77,756 GPU hours
>
> **Cost with Zest**
>
> 2040 hours (ZEST pretrain) + 0.075s (single iteration latency) * 52,000 (dataset size) * 3 (epochs) * 1000 (#num of tasks) = 34,500 GPU hours (2.25x less)
>
> The MORE models you will fine-tune, the MORE savings you will gain from ZEST.
>
> [1] https://crfm.stanford.edu/2023/03/13/alpaca.html

---

> > ### Author Response · Authors · 2023-11-22
> >
> > Dear reviewer,
> >
> > Thank you once again for providing high-quality reviews. As today is the final day for discussion, could you please let us know if our response has fully answered your questions? We would be more than happy to provide further details and responses for any additional questions you may have.
> >
> > Best regards,
> >
> > Authors

---

### Official Review · Reviewer_KcMr · 2023-11-05

**Soundness:** 3 good
**Presentation:** 3 good
**Contribution:** 3 good
**Rating:** 6
**Confidence:** 4

**Summary:**

This paper proposes a zeroshot sparse fine-tuning method that can achieve satisfactory results by only training a few layers. On both vision and language tasks, the proposed ZEST is effective and efficient.

**Strengths:**

1) The proposed zeroshot sparse fine-tuning method is intuitive and easy to apply. The whole pipeline is simple but effective.
2) ZEST proves effective on both visual and language tasks, which is inspiring and shows its generalization.
3) The resource optimization is obvious, which can greatly reduce the resource burden of fine-tuning.

**Weaknesses:**

1) The details of this paper need to be better presented. For example, the structure of the ZEST predictor is ambiguous. The pair-wise ranking loss is also not specified.
2) The experiment settings are not insufficient. The authors chose three backbone networks for the vision task, but all of them are CNNs. The transformer-based network is also needed to be proven effectiveness in vision task.
3) The dataset setting is simple. It will be better to try different dataset splits, especially reducing the datasets for constructing ZEST predictor.

**Questions:**

1) To construct the ZEST predictor, CIFAR-10, Aircraft, Cars, Flowers, Food, and Pets datasets are employed, while CIFAR-100 and CUB datasets are utilized for evaluation. It would be helpful to provide further details on the experimental setup, such as whether different dataset splits yield varying results.
2) During ZEST predictor training, two samples are inputted. It's worth investigating whether the number of samples plays a significant role in determining ZEST's performance.
3) What is the estimated time required for dataset construction, and is it justifiable to shift the time investment from fine-tuning to label collection? The decision may hinge on the model's ability to generalize across different dataset splits.

---

> ### Author Response · Authors · 2023-11-20
>
> We thank the reviewer for such detailed and valuable questions! Our point-to-point response is attached below.
>
> ### **Q1: ZEST Construction Time and Saving on Downstream Tasks**
>
> The construction and training of a ZEST predictor for vision tasks (ResNet-50) require 150 GPU hours, while language tasks (LLaMA-7B) necessitate 2040 GPU hours. It is important to note that this predictor pre-training is a **one-time process** and can be universally applied across various downstream tasks. Consider the goal of building 10,000 customized chatbots using datasets similar to Stanford Alpaca
>
> **Cost without Zest**
>
> 0.179s (single iteration latency) * 52,000 (dataset size) * 3 (epochs) * 10,000 (#num of tasks) = **77,756 GPU hours**
>
> **Cost with Zest**
>
> 2040 hours (ZEST pretrain) + 0.075s (single iteration latency) * 52,000 (dataset size) * 3 (epochs) * 1000 (#num of tasks) = **34,500 GPU hours (2.25x less)**
>
> The MORE models you will fine-tune, the MORE savings you will gain from ZEST.
>
> ### **Q2: Should Shift Investment from Fine-Tuning to Label Collection?**
>
> While we recognize that the quality and size of data are paramount in both pre-training and fine-tuning, obtaining the distribution of downstream datasets can be challenging or even impossible in certain cases, such as those involving privacy, security, and real-time applications. Consequently, collecting data beforehand becomes a formidable task.
>
> Moreover, GPUs offer a cost-effective and easily scalable solution. A 2000-GPU-hour job can be efficiently parallelized and completed on a cluster within a day. In contrast, data collection demands extensive engineering and labor efforts, incurring both expenses and a considerable amount of time.
>
> Moreover, the two methods can be combined and used together. For example, using ZEST to find important layers and then fine-tuning with quality-improved data.
>
> ### **Q3: Predictor Structure and Experiment Setup**
>
> The zest predictor is a feed forward network containing 4 feed-forward layers, with GELU as the activation layers.
>
> The pairwise loss computation involves two different batches rather than individual samples during training. Notably, our batch size is set to 64, and this parameter has a minimal impact on the predictor's accuracy during training, as shown below.
>
> | Resnet50 on CUB200 | Batch 64 | Batch 128 | Batch 256 |
> | --- | --- | --- | --- |
> | KD | 0.817 | 0.826 | 0.817 |
>
> In order to train the predictor, we carefully split the datasets on both vision and language tasks and ensure there is no overlap between. For vision, we use {CIFAR-10, Aircraft, Cars, Flowers, Food, and Pets} to construct the training set. For language models, we use {Arc-C, OpenbookQA, PIQA, and Sciq} datasets to train the predictor.
>
> ### **Q4: Different dataset splits when training ZEST predictors:**
>
> We have included additional ablation studies as to how the predictor’s performance changes as the number of training datasets change.
>
> | Resnet50 on Cifar100 | Predictor with 3 | Predictor with 6 | Predictor with 9 | Predictor with 12 | Predictor with 21 |
> | --- | --- | --- | --- | --- | --- |
> | ZEST @ 3 | 77.7 | 77.7 | 79.0 | 79.5 | 80.3 |
> | ZEST @ 5 | 80.0 | 80.0 | 80.1 | 80.3 | 80.7 |
> | ZEST @ 10 | 80.1 | 80.1 | 80.2 | 80.7 | 80.9 |
>
> The predictor with 3, 6, 9, 12 respectively mean the number of model/dataset combination used. 21 is the max number of model/datasets we have, namely 3 models on 7 datasets, excluding Cifar100. As we can see, the generalization ability to choose the best layers increases for the predictor as the number of training dataset/models are used.

---

> > ### Author Response · Authors · 2023-11-22
> >
> > Dear reviewer,
> >
> > Thank you once again for providing high-quality reviews. As today is the final day for discussion, could you please let us know if our response has fully answered your questions? We would be more than happy to provide further details and responses for any additional questions you may have.
> >
> > Best regards,
> >
> > Authors

---

### Official Review · Reviewer_XBCf · 2023-11-06

**Soundness:** 1 poor
**Presentation:** 2 fair
**Contribution:** 1 poor
**Rating:** 3
**Confidence:** 4

**Summary:**

This paper proposes a new surgical fine-tuning approach, where the authors train a predictor to estimate the importance of blocks given several proposed metrics computed from the mini-batch data. The metrics being considered are classified into static metrics, forward metrics and backward metrics. The authors assume it is possible to predict the layer contribution, i.e.,  the final task performance when fine-tuning only that single layer, given the mini-batch statistics only. Also, they assume such predictor could generalize to predict layer importance for unseen datasets. The authors evaluate ZEST on ResNet50, MobilenetV2, EfficientNet, BERT and LLAMA.

**Strengths:**

- The direction of considering the generalization of layer-wise surgical fine-tuning to deal with unseen datasets is a very appealing yet important task for machine leaning community.

- The paper is relatively easy to read and follow.

- The experiments consider both computer vision and NLP problems. Also, it is very interesting to see the popular large-scale model  LLAMA is considered for evaluating the sparse fine-tuning model.

**Weaknesses:**

My concerns on the paper center on the *novelty/soundness of the method*, the *comprehensiveness of the experimental results*, and the *weak reproducibility*, where **an appendix is missing** with my important details being not available to the readers.

- The novelty of this paper is rather limited. I feel the layer evaluation metrics, or so called zero-shot metrics in this paper, are the most important component for this type of works. However, regretfully **none of the metric introduced in this paper (see Sec 3.1) is new**. The metric formulas in Sec 3.1 have very high overlap with the reference paper Abdelfattah et al. 2021. Similarly, the layer contribution score is simply inferred from one layer/block fine-tuning accuracy, which is not novel either.

- The proposed method comes with a strong assumption, which is that the authors **assume the layer contribution score could be approximated with low computational cost by mini-batch data**. I would like to elaborate why such assumption is wrong. (1) First, it is straightforward that different mini-batch will result in highly biased and noisy metrics scores. This means for each dataset, there is no guarantee different mini-batch could result in stable and reliable evaluation statistics for each individual layer/block in deep neural networks. Even for the same mini-batch data, the zero-shot metrics (input to the ZEST predictor) keep changes as the model parameter is being fine-tuned. Do the authors collect such zero-shot metrics throughout model fine-tuning (inferenced under different $\Theta$s as the training data or collect that from the initial $\Theta$? (2) Second, for generalizing to testing datasets, the method would additionally assume the prediction model to be able to make prediction given the highly stochastic mini-batch statistics on unseen data. For secure generalization on unseen dataset, the authors adopt a very simple trick, which is to mix the data from different datasets in a minibatch. I am not convinced such trick could ensure the generalization ability of the predictor on such an unrealistic task on completely unseen dataset.  Overall, the task of estimating block/layer importance based on mini-batch data in the presented way fails to convince me.

- For all such layer-wise pruning methods, the most time consuming part is the data construction part. For this work, the authors adopt a non different data collection approach, where only one layer is fine-tuned to collect the $Score_{gt}$ (see **Training Label Collection** part in Figure 2).  If comparing the entire time from data collection to final model fine-tuning, I do not feel the total time efficiency is reduced in a great deal.

- In the abstract, the authors make very strong claim about the benefit of ZEST, saying their method **can decrease the number of trainable parameters by up to 99%, performing on par with full fine-tuning**. However, from the other part of this paper, it seems the sparsity of fine-tuning with ZEST is not as low and how the statistic of 99\% is derived is not explained from anywhere. I suggest the authors to carefully revise the writing of this paper to remove such overclaims.

- One major flaw of this paper is that for the main benchmark results (shown in Table 3), the authors did not explicitly specify which performance evaluation metric they use to make the comparison for BERT and LLAMA, while there could be many possibilities to consider. Also, the claim that **ZEST with 5 layer could outperform LLAMA's full fine-tuning by up to 5\%** is highly suspicious, because the reported performance score for LLAMA FT-Full on the *hellaswag* dataset is much lower than the publicly reported standards (e.g., refer to the scores for LLAMA-7b from https://github.com/ggerganov/llama.cpp/discussions/2321).

- Important details about the datasets and model training are missing. It is necessary to describe dataset statistics, model properties and training details apart from learning rate and mini-batch size.

- The author claim ZEST can be used orthogonal to the other method LoRA. However, adding LoRA to ZEST regretfully lowers the performance and does not bring positive effect (see Table 3). In this case, the conclusion seems to highlight the WEAKNESS or LIMITATION of ZEST, rather than its strength, diminishing the significance of this work.

- The experiment is not convincing also because that essential fine-tuning baselines are missing from the benchmark comparison. For the main results shown in Table 3, apart from ZEST variants, the authors only include two very simple fine-tuning baselines: **zero-shot** and **FT-Full**, which is insufficient. I believe there are many layer-wise deep neural net pruning methods that is related to this work. It is essential to include up-to-date SOTA fine-tuning baselines (e.g., [1]).

[1] AutoLR: Layer-wise Pruning and Auto-tuning of Learning Rates in Fine-tuning of Deep Networks (AAAI'21).

- Ablation study of ZEST is not comprehensive. I feel simply comparing with completely random baselines (e.g, Rand@3 and Rand@5 from Figure 3) is a bit unfair. It would be more interesting if random baselines which employ a combination of ZETA suggested layers and purely randomly chosen layers could be investigated.

- The authors focus on reporting the benchmark performance scores on the hold-out testing datasets, while the performance scores on the training datasets would be also interesting and important for the readers to know. I suggest the authors to add more comprehensive comparison results, with more inclusive datasets and related pruning baselines.

- The term **zero-shot metrics** sounds very confusing to me. From my understanding, those metrics, when being talked about for the training datasets, is not zero-shot at all. I feel it is not appropriate to name the metrics as zero-shot. Related work referenced paper, such as Abdelfattah et al. 2021, also does not use such term.

- From Fig 2, the prediction result of ZEST is fed to a **pair-wise ranking loss**. In the entire paper, such loss is not talked about at all. It is said the **$Score_{gt}[i]$** is given as the label for ZETA, with which supervised training can be done. Why additional ranking loss is there? Also, it is unclear why for **ZEST Inference**, no ranking information is inferenced and only **$Score_{gt}[i]$** is inferenced?

- The related work section is poorly written because it covers very limited literature on the most important layer-wise sparse fine-tuning methods, and the discussion on transfer learning seems too long. There are plenty of such works from the literature. The authors should discuss on them in detail.

**Questions:**

- From your algorithm (i.e., the **Sparse Fine-Tuning** part in Sec 3.2), you always use **top-n** layers with highest ZEST scores for fine-tuning. Does this mean no matter what testing datasets come in, you always predefine the number of layers as a hyperparameter? I wonder whether there is a way to verify the quality or sensitivity of ZEST prediction scores to identify optimal number of layers that can be customized to each testing dataset. From the training efficiency perspective, it seems not a good idea to always fix the number of n in your top-n fine-tuning strategy.

- In introduction, the authors claim **ZEST is 1000x more efficient than previous methods**. I wonder what is the 1000x in terms of? How exactly it is measured, e.g., how many datasets, compared to WHICH previous methods, is it for training or only inference, etc. In many other parts of this paper, there are also similar claims, e.g., 2.7x, 2.3x, for which, I think the authors should state more clearly how the numbers are derived.

---

> ### Author Response · Authors · 2023-11-21
>
> Thanks for your effort for reviewing and we appreciate your detailed comments. However, we would like to point out politely yet firmly point out that you might have some misunderstanding about our paper.
>
> - [**ZEST is not Pruning**] Our approach does not rely on pruning to maintain the integrity of the model architecture. Unlike pruning-based methods that may result in performance deterioration, **ZEST preserves the full architectures of the model**, ensuring flexibility and performance, as demonstrated in our experiments.
> - [**ZEST is a general method and can be plugged into many scenarios**] ZEST trains a general importance predictor that can adaptively select crucial layers for different downstream datasets. This offers a flexible and on-the-fly optimization approach.
>
> Before addressing the weaknesses and questions, let us provide a concise recap of the contributions of our method:
>
> - **[Generalization over different datasets] The conventional approach to predict layer importance relying on a single metric has shown limited promise, as demonstrated by the results in Table 1. In response, ZEST trains a predictor to generalize the selection of layer importance for different downstream tasks. The prediction shows a strong relevance with ground truth, making ZEST useful in practice.**
> - [**Thorough evaluation**] ZEST has been **thoroughly evaluated across both vision and language tasks**, confirming its level of generalizability. This empirical evidence strengthens the novelty of our approach in providing a more adaptive and broadly applicable solution.
> - **[Build one time, and deploy everywhere] ZEST only requires a single resource-intensive construction to train the predictor. For subsequent unseen downstream tasks, the ZEST predictor can be directly applied with a very low cost to select important layers for fine-tuning. This efficiency is particularly valuable in scenarios characterized by online data generation or personal customization where touching downstream data or labels is challenging.**
>
> Please check out the point to point response below.
>
> ### **Q1: Layer contribution score could be approximated with low computational metrics.**
>
> This is indeed a good question! It is true that different mini-batch will result in highly biased and noisy metrics scores, but the relevance with ground truth ranking (Section 4.2, Robustness of Single Zero-Shot Metrics) is actually consistent and stable. This inspired us directly to propose and design ZEST.
>
> To further clarify, we sample N (~20 - ~100) different minibatchs, and report the mean, variance, min, max, of the Kendall Tau, as shown below
>
> | Llama on hellaswag | Min | Max | Avg | Std Var |
> | --- | --- | --- | --- | --- |
> | |Grad| | 0.15 | 0.22 | 0.17 | 0.02 |
> | Snip | 0.20 | 0.27 | 0.23 | 0.04 |
> | Fisher | -0.01 | 0.09 | 0.03 | 0.04 |
> | Std. Wt | -0.61 | -0.07 | -0.50 | -0.07 |
>
> The results demonstrate that these zero-shot metrics **maintain stability:** variance with 100 minibatchs are small and the relative order keeps consistent. This affirms the reliability of the zero-shot metrics. This is further proved in our end2end experiments, where performance of ZEST is on par with that of fine-tune full, and consistently outperforms LoRA and random.
>
> ### **Q2: ZEST construction (pairwise loss) and inference (Score_GT) details**
>
> **Regression does NOT provide a comprehensive understanding of the layer's importance.** Our primary concern is layer rankings. The specific scores, whether relative, absolute, value, or magnitude, are not our main focus. Therefore, we utilize pair-wise training loss to train the ZEST predictor. Similar approaches have been employed in NAS, where the relative ranking of Neural Networks is more significant than their actual performance (e.g., [1],[2]). After obtaining the Score_Gt for each layer during inference, we sort them to determine the ranking order for each layer. Then, we can select the top-N layers for the next end-to-end fine-tuning.
>
> [1] Dynamic Ensemble of Low-fidelity Experts: Mitigating NAS “Cold-Start” (AAAI ‘23)
>
> [2] Evaluating Efficient Performance Estimators of Neural Architectures (NIPS ‘21)
>
> To further illustrate the advantage of pairwise loss, we conduct an ablation study to compare ZEST trained with Ranking-Loss and MSE below. The end2end fine-tuning accuracy is reported below.
>
> |  | Resnet50 |  | EfficientNet |  | Mobilenetv2 |  |
> | --- | --- | --- | --- | --- | --- | --- |
> |  | Cifar100 | Cub200 | Cifar100 | Cub200 | Cifar100 | Cub200 |
> | Best ZS | 0.31 | 0.25 | 0.66 | 0.40 | 0.38 | 0.68 |
> | ZEST w/ rank loss | 0.71 | 0.82 | 0.78 | 0.78 | 0.88 | 0.81 |
> | ZEST w/ MSE loss | 0.23 | -0.32 | -0.34 | -0.21 | 0.09 | -0.12 |
>
> We report Kendall Tau (the higher the better), evidently, ranking loss consistently demonstrates higher relevance than MSE.

---

> > ### Author Response · Authors · 2023-11-21
> >
> > ### **Q3: Is Top-N pre-determined for ZEST?**
> >
> > The Top-N is a flexible value and can be changed over different datasets. We follow current parameter-efficient fine-tuning methods (1% ~ 5% parameters) to choose top-3~10 layers to keep the same number of learnable parameters for a fair comparison.
> >
> > ### **Q4: Total time efficiency**
> >
> > For the ZEST predictor, it takes 150 GPU hours and 2040 GPU hours for vision (ResNet-50) and language tasks (Llama-7b) respectively. For single model training, ResNets and Llama-7B takes 0.5 and 1.5 hours respectively. However, we would like to emphasize that **such a predictor pre-training only needs to be performed once** and can be generally applied to various downstream tasks.  Considering building 10,000 customized chatbots using datasets similar as Stanford Alpaca[1]
> >
> > **Cost without Zest**
> >
> > 0.179s (single iteration latency) * 52,000 (dataset size) * 3 (epochs) * 10,000 (#num of tasks) = 77,756 GPU hours
> >
> > **Cost with Zest**
> >
> > 2040 hours (ZEST pretrain) + 0.075s (single iteration latency) * 52,000 (dataset size) * 3 (epochs) * 1000 (#num of tasks) = 34,500 GPU hours (2.25x less)
> >
> > The MORE models you will fine-tune, the MORE savings you will gain from ZEST.
> >
> > [1]: https://crfm.stanford.edu/2023/03/13/alpaca.html
> >
> > ### **Q5: ZEST can be used orthogonal to the other method PEFT methods such as LoRA**
> >
> > We would like to point out that although ZEST + LoRA decreases performance at 3,5 layers, fine-tuning only ~10 LoRA layers with ZEST can already perform on par with fully fine-tuning LoRA, while further decreasing the trainable parameter count.
> >
> > Considering multiple downstream tasks, this leads to numerous storage, for example 1000 tasks + 175B model, LoRA has to have many layers, but for ZEST + LoRA only changing 20% of the original LoRA layers is enough.
> >
> > To further elaborate out method’s compatibility with PEFT methods, we also conduct experiments with Adapter:
> >
> > | BERT | SST2 | MRPC |
> > | --- | --- | --- |
> > | FT-Full | 91.1 | 84.5 |
> > | Adapter | 90.5 | 82.8 |
> > | Adapter + ZEST @ 10 | 89.6 | 81.2 |
> > | Adapter + ZEST @ 15 | 90.1 | 82.6 |
> > | Adapter + ZEST @ 25 | 90.7 | 83.0 |
> >
> > ### **Q6: Clarification on ZEST savings**
> >
> > We thank the reviewer for reading to such detail, and we realize there is some misunderstanding on the savings reported.
> >
> > **Decrease the “trainable parameters by up to 99%”:**
> >
> > To clarify, this is the **saving ratio of trainable parameters** between ZEST and full fine-tuning. Using the example of LLaMA from Table 3, where the ratio is 90M/7B, resulting in 0.0128, the reduction effectively amounts to 99%.
> >
> > **Increase “efficiency by 1000X”:**
> >
> > To clarify, this is the saving **ratio** of **one** **forward backwards** pass ~60 GPU seconds for collecting zero-shot metric for ZEST, and the time for collecting the ground truth on LLaMA on Hellaswag is 224 layers * 1.5 Hours/layers = **336 GPU Hours**. Which is over 1000X speedup.
> >
> > **2.7x memory savings  2.3x speedup:**
> >
> > We refer the reviewer to table 4, where the inference latency and the memory speedup are listed. The detailed speedup calculation is (full fine-tuning memory/latency)/(ZEST fine-tuning memory/latency)
> >
> > ### **Q7: ZEST with 10 layer outperforms full fine-tuning / llama evaluation**
> >
> > The link reviewer reports the performance using `acc_norm`, while we report the “acc” instead. It is worth noting that  “acc_norm” is often (much) higher than “acc”, thus causing the misunderstanding and deviation from the link the reviewer provided.
> >
> > We choose to report top-1 acc for BERT and LLaMA to keep consistency. Specifically, we follow lm-eval[1] to perform evaluation to make the comparison fair.
> >
> > [1] [https://github.com/EleutherAI/lm-evaluation-harness](https://github.com/EleutherAI/lm-evaluation-harness)
> >
> > ### **Q8: Naming for zero-shot metrics**
> >
> > We take the naming convention from Neural Architecture Search (NAS), where in NAS such metrics are often called Zero-Shot metrics. Please let us known if you better naming suggestions.
> >
> > [1] Zero-Shot Neural Architecture Search: Challenges, Solutions, and Opportunities (IEEE TPAMI ‘23)
> >
> > [2] Evaluating Efficient Performance Estimators of Neural Architectures (NIPS ‘21)

---

> > > ### Author Response · Authors · 2023-11-21
> > >
> > > ### **Q9: Details about the datasets and model training**
> > >
> > > As for the dataset details, we would like to refer the reviewer to [GLUE](https://openreview.net/pdf?id=rJ4km2R5t7), and the LLaMA dataset benchmarks [Hellaswag](https://arxiv.org/abs/1905.07830), [Arc-E, Arc-C](https://arxiv.org/pdf/1803.05457.pdf), [OpenBookQA](https://arxiv.org/pdf/1809.02789.pdf), [sciq](https://www.semanticscholar.org/reader/932a5de79d8a8ebb75ea0c43493450fd9922e738), [PIQA](https://arxiv.org/abs/1911.11641).
> > >
> > > The training of the predictor as explained in Q2, we feed in batches of layer pairs, with zero-shot metric, and their relative ranks as output. For training the predictor we use a learning rate of 1e-4
> > >
> > > For fine-tuning, the settings for ZEST and Full fin-tuning are the same, as shown in the Setup section in 4.1. The learning rate is set to 2e−5 for LLaMA and 5e−5 for BERT. Additionally, a weight decay of 1e−2 is applied during training. Ensuring a consistent evaluation and comparison.
> > >
> > > ——————————————————————————————————————————
> > >
> > > We hope that our responses have addressed your concerns. Please let us know if there are any other experiments or clarifications we can provide to convince you to increase the rating.

---

> > > > ### Author Response · Authors · 2023-11-22
> > > >
> > > > Dear reviewer,
> > > >
> > > > Thank you once again for providing high-quality reviews. As today is the final day for discussion, could you please let us know if our response has fully answered your questions? We would be more than happy to provide further details and responses for any additional questions you may have.
> > > >
> > > > Best regards,
> > > >
> > > > Authors

---

### Meta-Review · Area_Chair_mDXb · 2023-12-07

**Metareview:**

This manuscript introduces an innovative surgical fine-tuning methodology wherein the authors deploy a predictor to gauge block importance through various metrics derived from mini-batch data. These metrics are categorized into static, forward, and backward metrics. The authors posit the feasibility of predicting layer contribution, specifically the ultimate task performance, solely based on mini-batch statistics during fine-tuning of an individual layer. Despite the potential significance of the proposed approach, the paper has sparked extensive deliberations among reviewers. Critiques primarily revolve around perceived inadequacies, including unsupported claims regarding the model's characteristics and concerns about reproducibility due to insufficient detail. Consequently, we contend that the present iteration of the manuscript requires further refinement before it can be deemed suitable for publication.

**Justification For Why Not Higher Score:**

N/A

**Justification For Why Not Lower Score:**

N/A

---

### Decision · Program_Chairs · 2024-01-16

Reject